# Comparative Analysis of Mesophilic YqfB-Type Amidohydrolases

**DOI:** 10.3390/biom12101492

**Published:** 2022-10-16

**Authors:** Roberta Statkevičiūtė, Mikas Sadauskas, Juta Rainytė, Karolina Kavaliauskaitė, Rolandas Meškys

**Affiliations:** 1Department of Molecular Microbiology and Biotechnology, Institute of Biochemistry, Life Sciences Center, Vilnius University, Sauletekio av. 7, 10257 Vilnius, Lithuania; 2Department of Eukaryote Gene Engineering, Institute of Biotechnology, Life Sciences Center, Vilnius University, Sauletekio av. 7, 10257 Vilnius, Lithuania

**Keywords:** ASCH, amidohydrolase, *N^4^*-acetylcytidine, prodrug

## Abstract

The widespread superfamily of the human activating signal cointegrator homology (ASCH) domain was identified almost 20 years ago; however, the amount of experimental data regarding the biological function of the domain is scarce. With this study, we aimed to determine the putative cellular functions of four hypothetical ASCH domain-containing amidohydrolase YqfB analogues by investigating their activity towards various *N*-acylated cytosine derivatives, including potential nucleoside-derived prodrugs, as well as their ability to bind/degrade nucleic acids in vitro. According to determined kinetic parameters, *N*^4^-acetylcytidine is assumed to be the primary substrate for amidohydrolases. Despite the similarity to the proteins containing the PUA domain, no nucleic acid binding activity was detected for YqfB-like proteins, suggesting that, in vivo, these enzymes are a part of the pyrimidine salvage pathway. We also demonstrate the possibility of the expression of YqfB-type amidohydrolases in both prokaryotic and eukaryotic hosts. The small protein size and remarkable halotolerance of YqfB-type amidohydrolases are of great interest for further fundamental research and biotechnological applications

## 1. Introduction

The human activating signal cointegrator-1 homology (ASCH) domain is widespread in all domains of life and some prokaryotic viruses [1]. It has been identified mainly at the C-terminus of ASC-1 (also known as thyroid receptor-interacting protein 4 (TRIP4)), which is associated with nuclear receptors and transcription factors and is part of an RNA-interacting complex [2]. ASCH superfamily proteins consist of ~110 amino acids and contain a conservative octapeptide G-X-K-X-X-[ETS]-X-R. ASCH domain-containing proteins have been shown to adopt a tertiary fold, which was originally observed in the RNA-binding pseudouridine synthase and archeosine transglycosylase (PUA) protein superfamily, even with low protein sequence similarity. Based on the known properties and functions of homologous domain-containing proteins in RNA metabolism, the ASCH domain is thought to be able to interact with RNA molecules as a transcription co-regulator [3,4]. However, there is a lack of available experimental data that highlight the biochemical characterization and biological functions of this diverse protein superfamily.

In a previous study, a hypothetical ASCH domain-containing protein from *Zymomonas mobilis* was described as a monomeric ribonuclease that binds nucleic acids and degrades single-stranded RNA molecules in vitro [5,6]. Recently, the *Escherichia coli* ASCH domain-containing protein YqfB was characterized as the smallest known amidohydrolase with only 103 amino acids [7]. The primary substrate of the enzyme is the modified nucleoside *N*^4^-acetylcytidine (ac4C), which is abundant in both eukaryotic and prokaryotic RNA molecules. The presence of ac4C is important for efficient protein synthesis and eukaryotic ribosome biogenesis [8,9,10,11,12]. The cytidine acetylation writer enzymes have been well studied in both prokaryotes and eukaryotes, but the prokaryotic erasers have not yet been identified.

Enzymes belonging to the large class of amidohydrolases are known to hydrolyze various linear and cyclic amides. These biocatalysts are attractive in biotechnology for their potential application in chemoenzymatic synthesis of pharmaceuticals. The amidohydrolase YqfB was shown to hydrolyze a well-known cancer prodrug, capecitabine, as well as *N*^4^-acylated 5-fluorocytosines (cytidines), suggesting potential application in a selective two-step activation system for the prodrug-targeted therapy.

Here we describe ASCH domain-containing proteins from mesophilic bacteria *Buttiauxella agrestis* (BagYqfB; 109 a.a.), *Cronobacter universalis* (CunYqfB; 113 a.a.), *Klebsiella pneumoniae* (KpnYqfB; 103 a.a.), and *Shewanella loihica* (SloYqfB; 107 a.a.). A comparative study showed that the recombinant proteins, including *E. coli* YqfB, were highly salt tolerant, and active towards a broad substrate range of *N*^4^-acylated cytosines and cytidines. In addition, we present data on the production of a functional recombinant YqfB from *E. coli* in the yeast *Pichia pastoris*.

## 2. Materials and Methods

### 2.1. Phylogenetic Analysis of ASCH Domain-Containing Proteins

In total, 1082 ASCH-domain-containing protein sequences were retrieved from the non-redundant protein sequences (nr) database using BLASTp search (https://blast.ncbi.nlm.nih.gov/Blast.cgi?PAGE=Proteins, accessed on 6 October 2022). The initial query was based on the YqfB amino acid sequence. The BLASTp search algorithm was set to exclude the *Escherichia* and *Klebsiella* genera, and to filter sequences outside the 50–80% sequence identity and 95–100% query coverage ranges. Multiple sequence alignment of these sequences, including the sequences of the YqfB and KpnYqfB, was performed with Clustal Omega [13] under default settings. The phylogenetic tree was visualized with iTOL v5 [14]

### 2.2. Strains and Chemicals

Cell cultures of the target microorganisms *Buttiauxella agrestis* DSM 9389, *Cronobacter universalis* DSM 27963, and *Shewanella loihica* DSM 17748 were obtained from the German Collection of Microorganisms and Cell Cultures (DSMZ). *Klebsiella pneumoniae* strain KV-3 is a veterinary isolate [15].

The *E. coli* DH5α (Novagen, Germany) strain was used for routine cloning experiments, while for the recombinant synthesis of target proteins, *E. coli* HMS174 *ΔpyrF ΔyqfB* cells were used. PichiaPink^TM^ expression system (Invitrogen, Carlsbad, CA, USA) strain 2, which is an *ade2* and *pep4* knockout, was used for YqfB expression in *Pichia pastoris*.

Cytidine, *N*^4^-acetylcytosine, *N*^4^-acetylcytidine, *N*^4^-acetyl-2′-deoxycytidine, and *N*^4^-isobutyryl-2′-deoxycytidine were purchased from Combi-Blocks (San Diego, CA, USA). *N*^4^-benzoylcytidine, capecitabine, 5′-fluorocytosine, and *p*-nitrophenyl acetate were obtained from Sigma Aldrich (St. Louis, MO, USA). Cytosine was purchased from Alfa Aesar (Germany), and isocytosine, 2′-deoxycytidine, and 5′-fluorocytidine were from Biosynth Carbosynth (Berkshire, UK).

The preparation of *N*^2^-acetylisocytosine [7] and *N*^4^-hexanoyl-2′-deoxycytidine, *N*^4^-nicotinoyl-2′-deoxycytidine, *N*^4^-(3-acetyl-benzoyl)-2′-deoxycytidine, and *N*^4^-(4-acetyl-benzoyl)-2′-deoxycytidine [16] was described in previous publications. The synthesis of other compounds is described in the Appendix A.

### 2.3. Growth Media

A standard Lennox LB medium was used for routine bacteria cultivation and protein synthesis induction (10 g/L tryptone, 5 g/L yeast extract, 5 g/L NaCl). SOB (20 g/L Bacto Tryptone, 5 g/L yeast extract, 0.5 g/L NaCl, 0.1 g/L KCl) was used for *E. coli* recovery after transformation. The complementation experiments with an auxotrophic *E. coli* HMS174 *ΔpyrF ΔyqfB* strain were carried out in a minimal M9 medium (3.5 g/L Na_2_HPO_4_, 1.5 g/L KH_2_PO_4_, 2.5 g/L NaCl, 0.2 g/L MgSO_4_, 0.01 g/L CaCl_2_) supplemented with 20 mg/L of ac4C.

For the cultivation of yeast cells, YPD (Yeast Peptone Dextrose; 1% yeast extract, 2% peptone, 2% dextrose) media with and without 2% agar was used for *P. pastoris* growth and storage. YPDS (YPD media with 1 M D-sorbitol) was used for *P. pastoris* cell recovery after transformation. PAD (Pichia Adenine Dropout media; 13.4% YNB-aa, 1.25% CMS-ADE, 0.005% biotin, 2% dextrose) with 2% agar was used for the growth and selection of *P. pastoris* transformants. BGMY (Buffered Glycerol-complex Medium; 1% yeast extract, 2% peptone, 100 mM potassium phosphate (pH 6.0), 1.34% YNB-aa, 0.00004% biotin and 1% glycerol) and BMMY (Buffered methanol-complex medium; 1% yeast extract, 2% peptone, 100 mM potassium phosphate (pH 6.0), 1.34% YNB-aa, 0.00004% biotin, and 0.5% methanol) was used for *P. pastoris* culture growth and induction.

### 2.4. Gene Cloning into Expression Vectors

Genes encoding ASCH domain-containing proteins were PCR-amplified using the 2× Phusion Polymerase Master Mix (Thermo Scientific, Vilnius, Lithuania) and cloned into a pLATE31 expression vector by aLICatorTM LIC Cloning and Expression System Kit 3 (Thermo Scientific, Vilnius, Lithuania). As a source of genomic DNA, intact cells of *B. agrestis*, *C. universalis*, *K. pneumoniae* KV-3, and *S. loihica* were used. Gene and primer sequences used in this study are listed in Appendix A. The cloning and expression procedures of the *E. coli yqfB* gene were described previously [7].

### 2.5. Purification of YqfB Analogues

A single colony of *E. coli* HMS174 *ΔpyrF ΔyqfB* cells transformed with the pLATE31 vector containing one of the target genes was inoculated into a 5 mL LB medium supplemented with ampicillin (50 µg/mL) and incubated at 37 °C with agitation (180 RPM) for 16 h. Overnight cultures were transferred into 50 mL of sterile LB medium and grown under the same conditions until OD_600_ reached 0.6–0.8. Protein synthesis was then induced by the addition of IPTG to a final concentration of 0.5 mM and incubated for an additional 12 h at 30 °C. The cells were harvested by centrifugation (10 min, 3220× *g*, 4 °C), resuspended in 20 mM Tris-HCl (pH 7.5) and 200 mM NaCl, and disrupted by sonication. The cell debris was removed by centrifugation (20 min, 16,000× *g*, 4 °C) and the supernatant was used for protein purification by the ÄKTA Pure protein purification system (Cytiva, Sweden). The cell-free extract was loaded on a 1 mL Ni^2+^-charged chelating column (Cytiva, Sweden) and, using a buffer containing 20 mM Tris-HCl (pH 7.5), 200 mM NaCl, and 500 mM imidazole, proteins were eluted by increasing the imidazole concentration. To remove the imidazole, protein samples were loaded on 2 × 5 mL desalting columns (Cytiva, Sweden) and eluted with 20 mM Tris-HCl (pH 7.5) and 200 mM NaCl buffer. Proteins were stored at −20 °C until usage. Protein purity was analyzed according to the standard Laemli SDS-PAGE protocol [17], then the concentration was measured using an Implen NanoPhotometer^®^.

### 2.6. Determination of Substrate Range

Hydrolysis products of 12 modified cytosine and cytidine derivatives were determined qualitatively by applying thin-layer chromatography (TLC). The reaction mixture consisted of 4 mM of modified cytidine or 8 mM of modified cytosine compounds, 20 mM Tris-HCl (pH 7.5), and 1 µM of protein of interest. After the incubation for 1 and 24 h in a 30 °C thermomixer, 1 µL of the reaction mixture was spotted on a silica gel coated aluminum plate (60 F254, Merck, Germany), placed into a TLC chamber with chloroform:methanol (5:1 *v*/*v*) as an eluent. The dried plates were visualized with short wave UV light (254 nm).

The reaction mixture with the *p*-nitrophenyl acetate consisted of 1 mM of *p*-NP acetate, 20 mM Tris-HCl (pH 7.5), and 1 µM of protein of interest. The absorption at 405 nm was measured spectrophotometrically after 30 min of incubation with an enzyme.

### 2.7. Enzyme Kinetics Assay

The activity of target amidohydrolases was measured by monitoring the hydrolysis of ac4C using the spectrophotometric method (Helios γ UV-Vis). The reaction mixture contained 20 mM Tris-HCl (pH 7.5), 0.025–0.2 mM of ac4C, and the appropriate amount of protein. Then, a decrease in absorbance at 310 nm was recorded for 30 s at 22 °C. The background hydrolysis of the substrate was determined by using a blank control with a composition identical to the reaction mixture, except the enzyme was replaced with a reaction buffer. All measurements were repeated at least three times. The kinetic parameters were determined by applying a non-linear regression model [18].

### 2.8. Relative Activity in Salts

The effect of salts on the enzyme activity was determined with NaCl and KCl concentrations of up to 4 M and 3 M, respectively. The reaction mixture contained 20 mM Tris-HCl (pH 7.5), 0.2 mM ac4C, and an appropriate amount of salt and protein. The hydrolytic activity was evaluated spectrophotometrically by measuring a decrease at A_310_. The measurements were taken at 22 °C for 30 s.

### 2.9. Electrophoretic Mobility Shift Assay (EMSA)

Lyophilized *E. coli* MRE 600 tRNA (Roche, Germany) was dissolved in ultrapure MilliQ water treated with diethylpyrocarbonate (DEPC). The 5′-ends of the tRNA molecules were labeled with [γ-^32^P]-ATP using T4 polynucleotide kinase (Thermo Scientific, Lithuania). The 5′-labeled RNAs were incubated with target proteins for 30 min at 37 °C in a binding mixture, containing 20 mM Tris-HCl (pH 7.5), 300 mM KCl, 10 mM MgCl_2_ or 10 mM CoCl_2_, and 100 ng/µL bovine serum albumin, similarly as described by Kim et al. [5]. The final ratio of tRNA to protein was 1:100. After the incubation, samples were loaded onto 8% (*v*/*v*) native polyacrylamide gel. Electrophoresis was conducted at 150 V for 150 min at 1× TBE buffer. The visualization was performed by phosphorimaging (Fuji, Japan).

### 2.10. Cloning and Expression of Recombinant YqfB in Pichia Pastoris

The *yqfB* gene was amplified by PCR using the pET21b-yqfB plasmid as a template with the following primers: YqfB_SmaI_For (5′-TCC**CCCGGG**ATGCAGCCAAACGACATCAC-3′) and YqfB_6xHis_KpnI_Rev (5′-GTCGGTA**GGTAC**CTTAGTGGTGATGGTGATGATGAAGACATTTAAATTCAATCAC-3′), where the highlighted parts are the restriction sites of SmaI and KpnI enzymes, respectively. The underlined part is the 6xHis-tag which is introduced at the C-terminus of YqfB. The resulting DNA fragment was cloned into the pPink-αF-HC (Invitrogen, Carlsbad, CA, USA) vector between the StuI and KpnI restriction sites. The constructed yeast expression vector pPink-αF-YqfB was verified by sequencing. 

Transformation of PichiaPinkTM strain 2 with pPink-αF-YqfB was performed according to the manufacturer’s recommendations via electroporation. The plasmid DNA was digested with the SpeI restriction enzyme and later extracted with the phenol/chloroform mixture for further use. Five micrograms of linearized plasmid were used for the transformation. PichiaPinkTM strains are *ade2* auxotrophs, so recombinant colonies were grown on PAD-Ag plates for an easy color-based selection of transformants. A single colony was inoculated in 12 mL of BGMY media in a 100 mL flask and grown at 30 °C in a shaking incubator for 2 days. The culture was used to inoculate 500 mL of BGMY media in 2 L baffled flasks and grown under the same conditions until OD_600_ reached 5–6. The cells were then harvested by centrifugation at 3000× *g* for 5 min. Protein expression was induced by resuspending the cells in 100 mL of BMMY media in 1 L baffled flasks. The culture was grown as described for another 72 h with the subsequent addition of methanol to a final concentration of 2% in 24-h intervals. The cells were pelleted by centrifugation at 3000× *g* for 20 min, and the supernatant was collected. For efficient protein purification from the *P. pastoris* culture medium, the collected supernatant was placed into a dialysis bag and dialyzed against a phosphate buffer solution (50 mM NaH_2_PO_4_, 300 mM NaCl, 10 mM imidazole; pH 8.0) at 4 °C for 48 h.

### 2.11. SDS-PAGE and Western Blot

Protein samples were mixed with a reducing sample buffer and heated to 95 °C prior to loading into a 14% SDS-polyacrylamide gel. Proteins were stained by the addition of Coomassie Brilliant Blue (Sigma-Aldrich Co.). After separation in SDS-PAGE, proteins were transferred to a polyvinylidene difluoride (PVDF) membrane (Cytiva, Amersham Pl, UK) by semi-dry electroblotting. The membrane was blocked with the Roti^®^Block protein-free blocking solution for 1 h (Carl Roth GmbH & Co. Kg, Karlsruhe, Germany), washed with PBS with 0.1% Tween-20 (PBS-T), and then incubated with primary 6xHis Tag Monoclonal antibodies (Mab) (HIS.H8) (Invitrogen, Carlsbad, CA, USA) 1:3000 diluted with PBS-T. After overnight incubation, the membrane was washed several times with PBS-T and subsequently incubated with goat anti-mouse IgG (H+L) HRP conjugate (Bio-Rad, Hercules, CA, USA) 1:3000 diluted with PBS-T. After washing, the chemiluminescence signal from the enzymatic reaction was developed by adding 4-chloro-1-naphthol and H_2_O_2_ (Fluka, Buchs, Switzerland) or 1-StepTM Ultra TMB-Blotting solution (Thermo Fisher Scientific, Vilnius, Lithuania).

### 2.12. Purification of YqfB from P. pastoris Growth Medium

Yeast-produced protein was purified by applying Ni^2+^ affinity chromatography. Dialyzed yeast culture media were loaded onto a Ni^2+^ immobilized column (Cytiva, Sweden), washed with a phosphate buffer solution, and eluted by increasing imidazole concentration. Then, gel filtration was applied to change the storage buffer to 20 mM Tris-HCl and 200 mM NaCl. The purified protein was concentrated using Amicon^®^ Ultra-4 Ultracel^®^—10K centrifugal filter unit (Millipore, Ireland).

### 2.13. Glycosylation Analysis

Recombinant protein glycosylation was analyzed by an adapted Western blot (WB) assay. Instead of the primary antibodies, the membrane was incubated with *Canavalia ensiformis* (Jack bean) lectin Concanavalin A peroxidase conjugate (Sigma-Aldrich Co.) diluted with PBS-T to a working concentration of 0.1 µg/mL. After overnight incubation, the membrane was washed two times with PBS-T and the enzymatic reaction product was visualized by adding the 1-StepTM Ultra TMB-Blotting solution (Thermo Scientific, Waltham, MA, USA).

## 3. Results

### 3.1. Amidohydrolytic Activity of YqfB Homologs

The potential proteins of interest for this study had to meet two main criteria: (1) sequence similarity to previously characterized YqfB from *E. coli* had to be from medium to high (between 50 and 90%), and (2) the availability of cultures in DMSZ and the possibility to grow them under available infrastructure. To do this, the phylogenetic tree was constructed using hits from the protein BLAST. Since *Klebsiella pneumoniae* isolate KV-3 (identity to YqfB—84%) was already present in the laboratory culture collection, the search was set to exclude *Escherichia* and *Klebsiella* genera from the initial query. Furthermore, the following parameters were applied: 50–80% sequence identity and query coverage of 95–100% to exclude truncated sequences. By performing this, in total, more than 1000 hits were obtained, meeting the abovementioned criteria. Based on the results of multiple sequence alignment, a phylogenetic tree was generated (Appendix A). For this study, four ASCH domain-containing proteins were chosen. These proteins were from mesophilic *γ*-proteobacteria *B. agrestis*, *C. universalis*, *K. pneumoniae*, and *S. loihica*, which fell into distinct phylogenetic groups based on the phylogenetic analysis results. Amplification of selected genes by PCR was followed by ligation independent cloning (LIC) into the pLATE31 expression vector. The amidohydrolytic activity of target proteins was first determined by transforming uridine auxotrophic *E. coli* HMS174 *ΔpyrFΔyqfB* cells with plasmids encoding YqfB-like proteins and growing the cells on minimal media supplemented with 20 µg/mL ac4C and 50 µg/mL ampicillin (Appendix A). After overnight incubation, only BagYqfB, CunYqfB, KpnYqfB, and SloYqfB-producing bacterial colonies appeared. Cells carrying pET21b-yqfB were used as a positive control, while cells transformed with the pET21b vector without an insert were used as a negative control. The ability of a uridine auxotrophic strain to grow on ac4C supplemented media was an indicator of the amidohydrolytic activity of the target protein [19].

Pairwise sequence alignment was used to compare the amino acid sequences of target enzymes with the previously described amidohydrolase YqfB. It was shown that KpnYqfB has the highest similarity (84%), followed by the less homologous BagYqfB (67%), CunYqfB (66%), and SloYqfB (52%) proteins. Additionally, the multiple sequence alignment (MSA) results showed that YqfB-like ASCH domain-containing proteins share identical amino acids at 39 positions. These include the catalytic residues Lys21, Thr24, Arg26, and Glu74 of YqfB (Appendix A). Sequence visualization showed that, as expected, most of the amino acids that make up the structure of the ASCH domain were conserved among the proteins studied, while the variable regions were located in loops and on the surface of the enzymes (Appendix A).

### 3.2. Substrate Specificity of Target Amidohydrolases

In order to qualitatively determine the substrate range of target proteins and to compare it with YqfB, purified enzymes were tested with *N*^4^- and *N^2^*-acylated cytosine as well as *N*^4^-acylated 5′-fluorocytosine derivatives (Table 1). Samples were analyzed by performing TLC after 1 and 24 h of incubation and YqfB was used as a comparative control for this assay. Hydrolysis of a chromogenic substrate, *p*-nitrophenyl acetate, was monitored spectrophotometrically at 405 nm.

Despite the differences in amino acid sequences, their substrate range was identical to YqfB: all target proteins were active towards *N*-acylated (5′-fluoro)cytosine compounds, harboring aliphatic, aromatic, and heterocyclic groups (Appendix A). Among 16 cytosine substrates tested, only two, *N*^2^-pivaloylisocytosine and *N*^2^-pivaloyl-5-fluoroisocytosine, were not hydrolyzed by target enzymes. This could be the result of the highly hydrophobic nature of the introduced modification. Furthermore, none of the enzymes were able to hydrolase the *p*-NP ester. It should be noted that all enzymes recognized *N*^4^-nicotinoyl-2′-deoxycytidine, *N*^4^-(3-acetyl-benzoyl)-2′-deoxycytidine, and *N*^4^-(4-acetyl-benzoyl)-2′-deoxycytidine; however, a significant amount of unhydrolyzed substrate was still detected even after 24 h incubation. 

Additionally, regarding the homology of ASCH domain-containing proteins to pseudouridine synthase and archeosine transglycosylase domain (PUA) proteins [3], together with the recently published data on ZmASCH nucleic acid-binding and ssRNA ribonucleolytic activity [5], the in vitro electromobility shift assay (EMSA) was performed using Ni-NTA purified target amidohydrolases (Appendix A). *E. coli* tRNA and synthetic ssRNA (17 and 30 nucleotides length), as well as ssDNA (17 nucleotides), were tested as substrates for YqfB-like amidohydrolases, in both the absence and presence of divalent metal ions (Mg^2+^ or Co^2+^; data not shown). PUA domain-containing tRNA pseudouridine synthase B (TruB) was used for positive RNA binding control (Figure 1).

The EMSA results revealed no nucleic acid binding or nuclease activity of the target proteins under the tested conditions. However, in the case of the homologous domain harboring TruB, used as a positive control in this experiment, the formation of protein-tRNA complexes was not dependent on metal ions. This indicates that RNA molecules are unlikely to be the native substrates of the tested YqfB analogues.

### 3.3. Catalytic Properties of YqfB Analogues

Based on the TLC results, the target proteins were most active towards ac4C as well as ac4dC (substrates were completely hydrolyzed within 1 h). Referring to the previously published YqfB data, we suggest that ac4C could be the primary substrate for the target proteins in the cell. Following this, the kinetics of the recombinant enzyme were analyzed (Table 2).

Similar kinetic parameters indicate that ac4C could be the primary substrate for all four investigated ASCH domain-containing proteins, which possessed a two-fold higher catalytic efficiency compared to YqfB. The largest difference appeared in the case of SloYqfB, which showed a lower affinity for the substrate, which was compensated for by the highest reaction rate among the tested enzymes. It should also be noted that a possible inhibition was not taken into account in the calculations. We assume that substrate hydrolysis occurs under the same acid–base mechanism as described for YqfB [7]. The unique catalytic triad, found at the domain’s conservative octapeptide, consists of Thr and Lys, which act as a nucleophile and a base, respectively, while Glu is an acid.

Unexpectedly, a high level of activity of the YqfB-like enzymes in high molarity of salts (NaCl, KCl) was determined (Figure 2).

All tested YqfB analogues showed a very high salt tolerance and only a slight (~20%) decrease in activity at extreme concentrations of KCl and NaCl was observed. From all tested proteins, KpnYqfB and CunYqfB possessed the highest and lowest tolerance to high salt concentrations, respectively, although the differences did not reach statistical significance.

### 3.4. YqfB Biosynthesis in a Yeast Expression System

Considering the possibility of using small amidohydrolases containing ASCH domains for therapeutic purposes, a GRAS-approved eukaryotic *P. pastoris* expression system was chosen as an alternative host. By adding a secretion sequence to YqfB, we were able to purify the enzyme from the culture medium. The SDS-PAGE results revealed two bands in the cell growth medium sample—the target YqfB band at ~12 kDa and another at ~15 kDa. After Ni-NTA purification, protein profile analysis revealed that the ~15 kDa protein elutes with YqfB (Figure 3a). The anti-6xHis Western blot analysis showed a target protein band at 12 kDa along with a low-intensity band at 15 kDa. This suggests that in yeast cells, a secretion sequence harboring YqfB could undergo post-translational modifications (Figure 3b).

To investigate, whether part of the bacterial protein is glycosylated in the yeast expression system, a Western blot with concanavalin A (ConA) was performed. ConA is known as a lectin that specifically binds α-D-mannosyl and α-D-glucosyl groups, and is used to detect glycosylated proteins [20]. The results showed a similar band at ~15 kDa, while the main portion of the protein was ConA negative (Figure 4b). YqfB amino acid sequence analysis revealed a single position where *N*-glycosylation could occur—Asn76 (see Appendix A) [21].

Qualitative enzyme activity experiments with ac4C showed that the yeast-produced protein retained its amidohydrolytic activity. The ability to produce a secreted active form of the target enzyme is important for a more convenient downstream process in biotechnology.

## 4. Discussion

In this work, we characterized four new ASCH domain-containing amidohydrolases from the mesophilic bacteria *B. agrestis*, *C. universalis*, *K. pneumoniae* KV-3, and *S. loihica*. The target proteins share moderate to high (52–84 %) sequence similarity to *E. coli* YqfB. Multiple sequence alignment revealed that catalytic Lys21, Thr24, and Glu74 amino acids, involved in the hydrolysis of modified cytidine compounds, are shared by all of the enzymes described in this study. According to the predicted 3D models, most of the conserved amino acids make up the core structure of the ASCH domain, while the variable regions are located in loops and on the enzyme surface. The above-mentioned catalytic triad found in BagASCH, CunASCH, KpnASCH, and SloASCH along with YqfB is an atypical and unique feature of this group of amidohydrolases [7]. The enzymes exhibited broad substrate specificity with *N*-acylated cytosine compounds. Based on the available enzymatic reaction parameters, we assume that in the cell, the main substrate of these enzymes is ac4C, since the calculated catalytic efficiency (k_cat_/K_M_) of all target enzymes is ~4.5 × 10^6^. In addition to their small size and high catalytic activity, the target proteins retain ~80 % of their activity in 3 M KCl and 4 M NaCl. The significant halotolerance of these small amidohydrolases could be the result of an acidic amino acid-rich protein sequence [22,23]. For the studied YqfB analogues, Glu and Asp residues constitute up to 20 % of the amino acids in the sequence. 

It is known that the number of ac4C positions in RNA molecules increases when the cell is exposed to environmental stress, which would suggest that the introduction of this modification is another example of dynamic adaptation [24,25]. However, there is no evidence for the reversibility of this modification in the bacteria domain. The high specificity of YqfB and its analogues to ac4C as well as the published data on ZmASCH ribonuclease activity have led to the hypothesis that these small amidohydrolases could carry out RNA deacetylation in vivo or be involved in the degradation of acetylated RNAs [5]. However, since no nucleic acid binding or ribonuclease activity was detected, the high catalytic activity of the tested enzymes suggests that free acetylated cytidine nucleotides are the cellular substrates rather than RNAs. This points to a possible role of ASCH domain-containing proteins in cytidine metabolism, since removing the acyl-group from modified cytidine could return the nucleotide to the pyrimidine salvage pathway. It is also possible that RNA-YqfB binding in vivo requires additional docking proteins, which were not analyzed in this study.

The ability of YqfB-like proteins to remove cytidine modifications is not only of fundamental importance but also of great interest for practical applications. In this study, we showed that all target proteins are capable of hydrolyzing capecitabine and other 5-fluorouracil-derived compounds. This suggests the possible role of ASCH domain-containing proteins in the prodrug activating systems, where an inactive prodrug is enzymatically converted into a cancer cells-targeting molecule. Considering the halotolerance of the YqfB-like enzymes, increased stability of these proteins in the blood could also be suggested. Although the bacterial expression system is efficient for the biosynthesis of these small amidohydrolases, a GRAS organism should be considered for potential application in pharmaceuticals. Here we show that the addition of a secretion signal sequence to the *N*-terminus of *E. coli* YqfB led to successful functional enzyme synthesis and transport to the extracellular environment by employing a well-studied eukaryotic expression system, *P. pastoris,* thus expanding the potential applications of these small amidohydrolases.

## Figures and Tables

**Figure 1 biomolecules-12-01492-f001:**
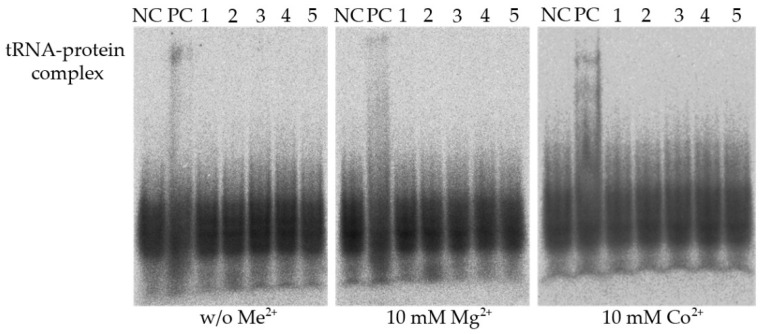
Evaluation of binding of YqfB and ASCH domain-containing YqfB-like proteins to tRNA in the presence and absence of metal ions in EMSA. NC—tRNA control without added protein samples, PC—positive control, tRNA pseudouridine synthase B (TruB). 1-5—tRNA incubated with YqfB, BagYqfB, CunYqfB, KpnYqfB and SloYqfB, respectively.

**Figure 2 biomolecules-12-01492-f002:**
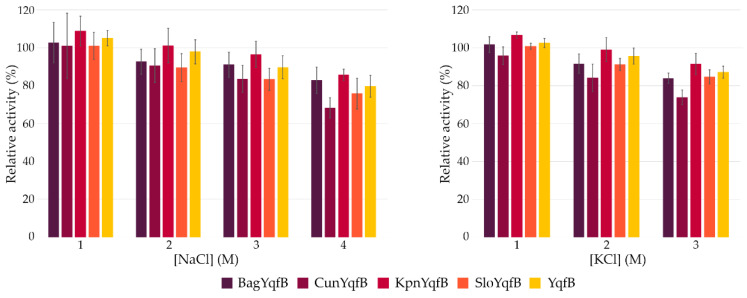
Effect of NaCl and KCl on the activity of ASCH domain-containing proteins. The activity without added salts was taken as 100%. The substrate used in these experiments was ac4C.

**Figure 3 biomolecules-12-01492-f003:**
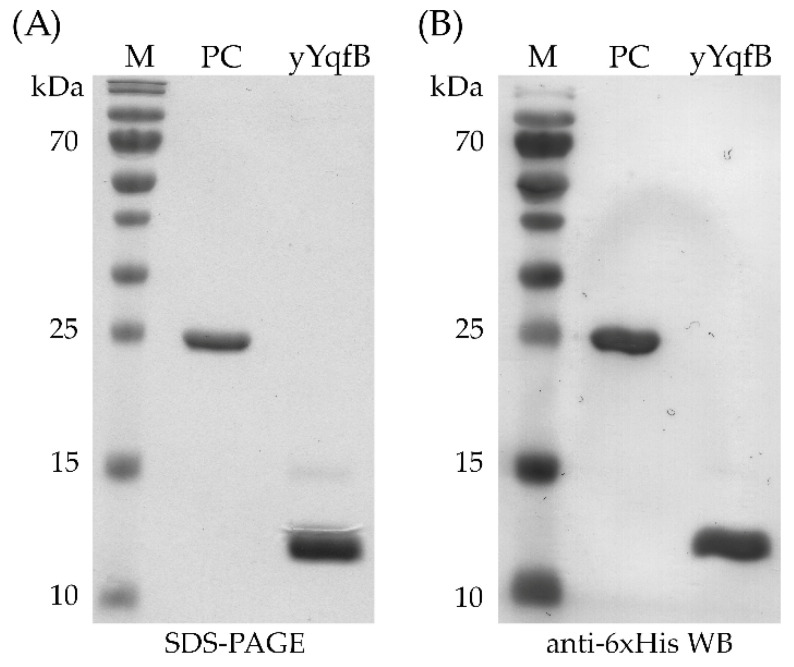
Analysis of YqfB produced in yeast *P. pastoris* (yYqfB). (**A**) Protein profile in SDS-PAGE, where M is the PageRuler Prestained Protein Ladder; PC is the 6xHis-tagged *Penaeus monodon* allergen Pen m4 –positive control for anti-6xHis Western blot analysis; yYqfB is the yeast produced *E. coli* YqfB; (**B**) Western blot analysis of described samples.

**Figure 4 biomolecules-12-01492-f004:**
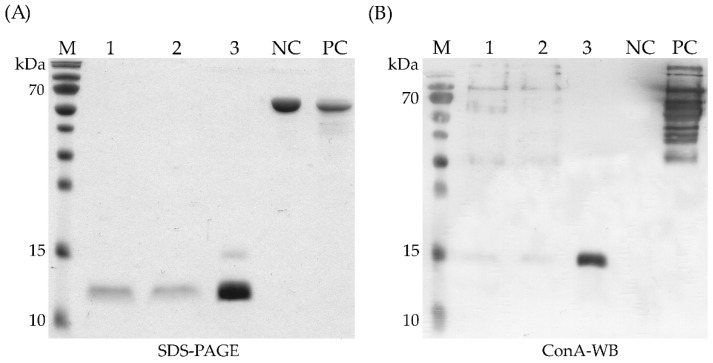
Determination of *N*-glycosylation of YqfB after expression in *P. pastoris* cells. (**A**) Protein profiles in SDS-PAGE and (**B**) Western Blot analysis with ConA. 1–3—protein samples prior to (1) and after (2) dialysis of yeast growth media and purification by affinity chromatography (3). PC—*P. pastoris*-produced glycosylated black tiger shrimp protein, Pen m 4, fused to the Maltose-binding protein (MBP), acts as a positive control for Western blot analysis with ConA. NC—*E. coli*-produced Pen m 4 protein fused to MBP acts as a negative control.

**Table 1 biomolecules-12-01492-t001:** A list of compounds used for qualitative determination of enzyme-substrate specificity. “+” indicates substrate hydrolysis within 24 h of incubation, “−”—hydrolysis did not occur.

Substrate	YqfB	BagYqfB	CunYqfB	KpnYqfB	SloYqfB
*N*^4^-acetylcytosine	+	+	+	+	+
*N*^4^-acetylcytidine	+	+	+	+	+
*N*^4^-benzoylcytidine	+	+	+	+	+
*N*^4^-acetyl-2′-deoxycytidine	+	+	+	+	+
*N*^4^-isobutyryl-2′-deoxycytidine	+	+	+	+	+
*N*^4^-hexanoyl-2′-deoxycytidine	+	+	+	+	+
*N*^4^-nicotinoyl-2′-deoxycytidine	+	+	+	+	+
*N*^4^-(3-acetyl-benzoyl)-2′-deoxycytidine	+	+	+	+	+
*N*^4^-(4-acetyl-benzoyl)-2′-deoxycytidine	+	+	+	+	+
*N*^2^-acetylisocytosine	+	+	+	+	+
*N*^2^-pivaloylisocytosine	−	−	−	−	−
Capecitabine	+	+	+	+	+
*N*^2^-acetyl-5-fluoroisocytosine	+	+	+	+	+
*N*^4^-benzoyl-5-fluorocytidine	+	+	+	+	+
*N*^4^-pivaloyl-5-fluorocytidine	+	+	+	+	+
*N*^2^-pivaloyl-5-fluoroisocytosine	−	−	−	−	−
*p*-nitrophenyl acetate	−	−	−	−	−

**Table 2 biomolecules-12-01492-t002:** Kinetic parameters of YqfB and ASCH domain-containing YqfB-like proteins. The activity was measured in 20 mM Tris-HCl, pH 7.5, at 22 °C, using ac4C as a substrate.

Enzyme	K_M_ (M)	k_cat_ (s^−1^)	k_cat_/K_M_ (M^−1^s^−1^)
YqfB [7]	(6.2 ± 0.1) × 10^−5^	157 ± 1	(2.2 ± 0.1) × 10^6^
BagYqfB	(3.6 ± 0.8) × 10^−^^5^	154 ± 11	(4.3 ± 0.7) × 10^6^
CunYqfB	(5.7 ± 0.8) × 10^−5^	248 ± 20	(4.4 ± 0.5) × 10^6^
KpnYqfB	(4.6 ± 0.8) × 10^−5^	216 ± 2	(4.8 ± 0.8) × 10^6^
SloYqfB	(1 ± 0.2) × 10^−4^	441 ± 37	(4.3 ± 0.5) × 10^6^

## Data Availability

Not applicable.

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
