# Peer review of "Comparative Analysis of Mesophilic YqfB-Type Amidohydrolases"

_biomolecules, 2022, doi:10.3390/biom12101492_

Round 1

Reviewer 1 Report

In their manuscript Statkevičiūtė et al. describe the comparative analyses of four YqfB-type amidohydrolases. These enzymes possess the same substrate specificity as YgfB, which was characterized by the authors in a previous study (Sci Rep 2020, 10, 788).

Overall, the manuscript is nicely written and the experiments are described in sufficient detail to allow their reproduction. However, I am concerned that the article does not really include much new insights into the YgfB family of proteins. Moreover, there a couple of issues, which need to be addressed before the manuscript becomes acceptable for publication:

1) The authors should comment on why they chose the YqfB homologs from Buttiauxella agrestis, Cronobacter universalis, etc. for their analysis. Otherwise, the study is somewhat arbitrary.

2) What do we learn from the pairwise sequence alignment of the YgfB homologs except that the catalytic residues are conserved? - Please explain.

3) It is a little bit strange that the catalytic efficiencies of BagYgfB, CunYgfB, KpnYgfB and SloYgfB are in the same range and that all of them are superior to the previously characterized YqfB (Table 2). Do the authors have a plausible hypothesis, which can explain these results? The difference in catalytic efficiency definitely deserves a much more thorough discussion.

4) I honestly do not see a good reason for expressing a prokaryotic protein in Pichia pastoris. Why did you not choose a bacterial host with GRAS status? In this way, you would not need to worry about protein glycosylation

Minor comments:

Figures 1 and 2 should be moved into the Supporting Information

Author Response

Dear Reviewer,

we are thankful for your critical and comprehensive analysis of the manuscript and constructive suggestions on how to improve it. You will find a point-by-point response in the attachment.

Roberta Statkevičiūtė
Department of Molecular Microbiology and Biotechnology, 
Institute of Biochemistry, Life Sciences Center, 
Vilnius University, Lithuania

Reviewer 2 Report

In this manuscript the authors have partially characterized YqfB homologs with respect to substrate specificity.  They identified 4 bacterial YqfB homologs and have shown that each of these proteins can catalyze the hydrolysis of acetyl groups from either cystosine (or cytidine) or isocytosine substrates.  Detailed kinetic constants were obtained with all five enzymes using N4-acetyl cytosine as the primary substrate.  They also demonstrated that modified cystosine on RNA were not substrates and thus it would appear that this enzyme activity is restricted to the salvage of modified cytidine or cytosine substrates.  

It would have been of interest to the readers of this article to know whether this enzyme can hydrolyze acetyl groups from acetylated guanine or adenine molecules. 

amidohydrolase is misspelled on line 362

Author Response

Dear Reviewer,

we would like to thank you for a comprehensive analysis of the manuscript and suggestions on how to improve it.

Our aim was to investigate the potential cellular function of YqfB-type amidohydrolases. Based on this, the main substrate was N4-acetylcytidine, which is a common and naturally occurring modification of coding and non-coding RNA molecules. However, the ability of our target enzymes to hydrolyse alternative substrates is of great fundamental interest. While searching for possible substrates in RNA modification database MODOMICS (https://doi.org/10.1093/nar/gkab1083), we found a single example of acetylation other than ac4C. N6-acetyladenosine (ac6A) was found on the bulk tRNA from Methanopyrus kandleri. The rare occurrence of this modification limits the availability of such substrate, hence, further studies are needed to elucidate a substrate range of YqfB-related amidohydrolases.

Roberta Statkevičiūtė
Department of Molecular Microbiology and Biotechnology, 
Institute of Biochemistry, Life Sciences Center, 
Vilnius University, Lithuania

Reviewer 3 Report

In this study, Statkevičiūtė and colleagues question the activity of four hypothetical analogues of the amidhohydrolase YqfB against a battery of N-acylated cytosine derivatives and their ability to bind nucleic acids in vitro.

The results convincingly demonstrate the broad substrate specificity of all four proteins with N-acylated cytsosine compounds, and also that the enzymes lack affinity for nucleic acids under the conditions tested.  The study also demonstrates the extreme tolerance of the enzymes to high salt concentrations, and reports the production of one of the enzymes in a yeast expression system for future biomedical applications.    

The manuscript is concise and well written and I recommend its publication without further changes.

The authors may want to consider adding a (supplementary) figure with the drawing of the different substrates listed in Table 1 to make it easier for those less familiar with these compounds to read

Author Response

Dear Reviewer,

we would like to thank you for a comprehensive analysis of the manuscript and suggestions on how to improve it. All the structures of used substrates can be found in the supplementary information Figure S5.

Roberta Statkevičiūtė
Department of Molecular Microbiology and Biotechnology, 
Institute of Biochemistry, Life Sciences Center, 
Vilnius University, Lithuania

Round 2

Reviewer 1 Report

My questions have been satisfactorily addressed by the authors.